

# Association of the CACNA2D2 gene with schizophrenia in Chinese Han population

Yingli Fu[1,2], Na Zhou[3], Wei Bai[1], Yaoyao Sun[1], Xin Chen[1],
Yueying Wang[1], Mingyuan Zhang[1], Changgui Kou[1], Yaqin Yu[1] and
Qiong Yu[1]

[1] Department of Epidemiology and Statistics, School of Public Health, Jilin University, Changchun, Jilin, China
[2] Division of Clinical Research, First Hospital of Jilin University, Changchun, Jilin, China
[3] Department of Pharmacy, School and Hospital of Stomatology, Jilin University, Changchun, Jilin, China

Corresponding author
Qiong Yu, yuqiong@jlu.edu.cn

## ABSTRACT

**Background:** Schizophrenia (SCZ) is a severely complex psychiatric disorder in which ~80% can be explained by genetic factors. Single nucleotide polymorphisms (SNPs) in calcium channel genes are potential genetic risk factors for a spectrum of psychiatric disorders including SCZ. This study evaluated the association between SNPs in the voltage-gated calcium channel auxiliary subunit alpha2delta 2 gene (*CACNA2D2*) and SCZ in the Han Chinese population of Northeast China.
**Methods:** A total of 761 SCZ patients and 775 healthy controls were involved in this case-control study. Three SNPs (rs3806706, rs45536634 and rs12496815) of *CACNA2D2* were genotyped by the MALDI-TOF-MS technology. Genotype distribution and allele frequency differences between cases and controls were tested by Chi-square ($\chi^2$) in males and females respectively using SPSS 24.0 software. Linkage disequilibrium and haplotype analyses were conducted using Haploview4.2. The false discovery rate correction was utilized to control for Type I error by R3.2.3.
**Results:** There was a significant difference in allele frequencies ($\chi^2$ = 9.545, $P_{adj}$ = 0.006) and genotype distributions ($\chi^2$ = 9.275, $P_{adj}$ = 0.006) of rs45536634 between female SCZ patients and female healthy controls after adjusting for multiple comparisons. Minor allele A (OR = 1.871, 95% CI [1.251–2.798]) and genotype GA + AA (OR = 1.931, 95% CI [1.259–2.963]) were associated with an increased risk of SCZ. Subjects with haplotype AG consisting of rs45536634 and rs12496815 alleles had a higher risk of SCZ (OR = 1.91, 95% CI [1.26–2.90]) compared those with other haplotypes.
**Conclusions:** This study provides evidence that *CACNA2D2* polymorphisms may influence the susceptibility to SCZ in Han Chinese women.

## INTRODUCTION

Schizophrenia (SCZ) is a severely debilitating psychiatric disorder characterized by positive and negative symptoms as well as cognitive dysfunction (*Allen et al., 2008*; *Koike et al., 2014*). The lifetime risk of SCZ is approximately 1% across the world (*Mayilyan, Weinberger & Sim, 2008*), and the lifetime prevalence of adults in China was 0.6% (*Huang et al., 2019*). SCZ has a marked impact on the life quality and accounts for

approximately 2.8% of the global burden of diseases reported by the World Health Organization in 2001, and the prevalence of SCZ disability was 0.41% in China (*Liu et al., 2015*).

Genetic and environmental factors may combine to increase SCZ risk (*Plomin, Owen & McGuffin, 1994*). Genetic factors explain ~80% of the risk for SCZ, and the risk of the SCZ decrease as the parental relationship recedes (*Zhu et al., 2009*). Despite vigorous genome-wide association studies have been conducted to elucidate the common genetic variations associated with the susceptibility to SCZ (*Børglum et al., 2014*; *Riley et al., 2010*), the etiology of SCZ remains obscure, suggesting that additional studies are required to discover the "missing heritability."

The voltage-gated calcium channel auxiliary subunit alpha2delta 2 gene (*CACNA2D2*), located in 3p21.31 and highly expressed in the brain, encodes a calcium channel protein. Voltage-gated calcium channels are widely distributed throughout the brain and mediate the intracellular Ca2+ influx of synaptic action potentials (*Guan et al., 2016*). In recent years, the gene encoding voltage-gated calcium channel subunit attracts wide attention in the field of SCZ pathogenesis. SNPs in calcium channel genes have been identified as genetic risk factors for a spectrum of psychiatric disorders (*Cross-Disorder Group of the Psychiatric Genomics Consortium, 2013*; *Cross-Disorder Group of the Psychiatric Genomics Consortium, 2014*). For example, several studies showed that SNPs of *CACNA1C* were significantly associated with SCZ, and the finding has been confirmed in different populations (*Gassó et al., 2016*; *Zhang et al., 2017*; *Zhu & Li, 2019*). Although there are no reported studies on the relationship between *CACNA2D2* and SCZ, *CACNA2D2* may cause other psychiatric disorders (*Berridge, 2013*) and severe neurological diseases (*Strupp et al., 2005*). In this study, we conducted a genetic association study to examine the association between SNPs of *CACNA2D2* and SCZ by a case-control study.

## MATERIALS AND METHODS

### Study population

A total of 761 patients with SCZ from the Mental Hospital of Changchun and 775 healthy controls from the physical examination center of the First Hospital of Jilin University were recruited. All participants were Han Chinese. The patients were diagnosed according to the criteria of the International Statistical Classification of Disease and Related Health Problems, Tenth Revision (ICD-10) independently by at least two experienced psychiatrists. Healthy controls matched the patients by gender and age. All controls had no personal or family history of mental illness. This study was performed under protocols approved by the Ethics Committee of Jilin University, China (2014-05-01). All subjects signed written informed consent before participating in this study, and all experiments were performed in accordance with relevant guidelines and regulations.

### DNA extraction and SNP selection

Peripheral blood of five mL was collected from each subject, and the genomic DNA was extracted from blood samples using a commercial DNA extraction kit (Kangwei Biotech Company, Beijing, China). SNPs located in the promoter and 3′ untranslated region

**Table 1 Primers for polymerase chain reaction.**

| SNP | Primer sequence (5′–3′) |
|---|---|
| rs12496815 | F: ACGTTGGATGTGGTTTTGGCACCAGTGCGT |
| | R: ACGTTGGATGTGGCACCCAAATCACATCTC |
| rs3806706 | F: ACGTTGGATGTGAGCTCAACAGCTGCCTTC |
| | R: ACGTTGGATGGTCCAGCAAACAGGTAAGAG |
| rs45536634 | F: ACGTTGGATGCAATGTATGTCAAGGGCCTG |
| | R: ACGTTGGATGGAGTCCCACTTAGTGCTCTG |

(UTR) of *CACNA2D2* were searched in NCBI-SNP (https://www.ncbi.nlm.nih.gov/snp/) and Ensembl (https://asia.ensembl.org/) databases. We predicted the function of these SNPs in SNPinfo (https://snpinfo.niehs.nih.gov/) and searched for minor allele frequency (MAF) each SNP in 1,000 Genomes. Finally, we chose three SNPs (MAF > 0.05) located in promoter or 3′UTR regions (rs3806706 in the promoter region and rs45536634 and rs12496815 in the 3′ UTR) of *CACNA2D2* and predicted to be located in transcription factor binding sites. SNP genotyping was performed using matrix-assisted laser desorption/ionization time of flight mass spectrometry (MALDI-TOF-MS). SNP genotyping reactions were performed in a 384-well Spectro-CHIP using a Mass Array nanodispenser (Sequenom Inc., San Diego, CA, USA). The primers for genotyping were designed by AssayDesigner3.1 and were listed in Table 1.

### Statistical analysis

Pearson's Chi-square ($\chi^2$) test and Student's *t*-test were used to test the distribution of sex and age between case and control groups, respectively. The distributions of allele and genotype were analyzed using $\chi^2$ tests. The odds ratio (OR) was used to estimate the relative risk of SCZ associated with genotypes with minor alleles. The Type I error due to multiple testing was corrected by the FDR method. All the above analyses were performed using Software SPSS 24.0 (IBM SPSS, IBM Corp, Armonk, NY, USA), and R version 3.2.3 was used for FDR corrections. The Hardy–Weinberg equilibrium (HWE) test was conducted in the case and control group separately by Goodness of fit $\chi^2$ test using online software SNPStats (https://www.snpstats.net/snpstats/start.htm). The linkage disequilibrium (LD) between SNPs was estimated in both females and males separately using Haploview 4.2 (*Barrett et al., 2005*), and the haplotype analysis was further performed using Haploview. The statistical power for each SNP was calculated according to the MAF of each SNP (rs45536634: 0.073, rs3806706: 0.378 and rs12496815: 0.388). The prevalence of SCZ (1%) was estimated by Quanto 1.2.4 (*Gauderman, 2002*). The OR was set from 1.4 to 2.0. All tests were two-sided and $P_{adj}$-value less than 0.05 was considered to be statistically significant.

## RESULTS

### Demographic characteristics

A total of 1,536 subjects were included in this study, comprised of 761 SCZ patients (58.2% males, Mean age = 34.61 ± 12.02 years) and 775 healthy controls (56.2% males,

**Table 2 Test of HWE for case and control groups.**

| Gene | SNP | Case | | | | Control | | | |
|------|-----|------|------|----------|------|---------|------|----------|------|
| | | Ho | He | $\chi^2$ | P | Ho | He | $\chi^2$ | P |
| CACNA2D2 | rs45536634 | 0.172 | 0.171 | 0.024 | 0.9 | 0.127 | 0.137 | 2.42 | 0.12 |
| | rs12496815 | 0.469 | 0.497 | 2.126 | 0.1 | 0.498 | 0.499 | 0.002 | 0.968 |
| | rs3806706 | 0.458 | 0.437 | 1.66 | 0.2 | 0.403 | 0.422 | 1.583 | 0.208 |

Note:
Ho, observed heterozygosity; He, expected heterozygosity.

**Table 3 Genotype and allele distributions of CACNA2D2 SNPs in female.**

| SNPs | Genotype | Case | Control | $\chi^2$ | P | $P_{adj}$ | OR [95% CI] |
|------|----------|------|---------|----------|------|-----------|-------------|
| rs3806706 | GG | 140 | 158 | 0.032 | 0.858 | 0.858 | 1 |
| | GC + CC | 165 | 181 | | | | 1.184 [0.689–2.035] |
| | Allele | | | | | | |
| | G | 417 | 460 | 0.039 | 0.843 | 0.858 | 1 |
| | C | 193 | 218 | | | | 0.977 [0.772–1.235] |
| rs45536634 | GG | 227 | 297 | 9.275 | 0.002 | 0.006* | 1 |
| | GA + AA | 62 | 42 | | | | 1.931 [1.259–2.963]* |
| | Allele | | | 9.545 | 0.002 | 0.006* | |
| | G | 513 | 635 | | | | 1 |
| | A | 65 | 43 | | | | 1.871 [1.251–2.798]* |
| rs12496815 | GG | 62 | 69 | 1.421 | 0.233 | 0.395 | 1 |
| | GA + AA | 190 | 268 | | | | 0.789 [0.534–1.165] |
| | Allele | | | | | | |
| | G | 247 | 308 | 1.268 | 0.263 | 0.395 | 1 |
| | A | 257 | 366 | | | | 0.876 [0.695–1.103] |

Notes:
$P_{adj}$ represent P corrected by FDR.
* $P_{adj} < 0.05$.

Mean age = 34.74 ± 11.41 years). There was no significant difference either in sex ($\chi^2 = 0.681$, $P = 0.409$) or age ($t = 0.221$, $P = 0.825$) between patients and healthy controls. The results of the HWE test were shown in Table 2. All SNPs were in accordance with the HWE in both cases and controls ($P > 0.05$).

## The distribution of alleles and genotypes in males and females

The detection rate of rs45536634, rs12496815 and rs3806706 were 97%, 92% and 98%, respectively. Genotype and allele frequencies of females were depicted in Table 3. A significant difference ($P_{ajd} = 0.012$) was observed in allele frequencies of rs45536634 between female SCZ patients and female healthy controls. Subjects who carried minor allele A had a 1.9 times higher risk of SCZ than those homozygous for the major G allele. Similarly, a significant difference ($P_{ajd} = 0.006$) was observed in the genotype distribution in females, and subjects with the minor allele (GA + AA) has an increased risk of SCZ when compared those with genotype GG (OR = 1.931, 95% CI [1.259–2.963]).

**Table 4 Genotype and allele distributions of *CACNA2D2* SNPs in male.**

| SNPs | Genotype | Case | Control | $\chi^2$ | P | $P_{adj}$ | OR [95% CI] |
|---|---|---|---|---|---|---|---|
| rs3806706 | GG | 190 | 225 | 5.373 | 0.02 | 0.12 | 1 |
| | GC + CC | 241 | 208 | | | | 1.372 [1.050–1.793] |
| | Allele | | | | | | |
| | G | 580 | 617 | 3.185 | 0.074 | 0.172 | 1 |
| | C | 282 | 249 | | | | 1.205 [0.982–1.478] |
| rs45536634 | GG | 361 | 369 | 0.127 | 0.721 | 0.865 | 1 |
| | GA + AA | 67 | 64 | | | | 1.070 [0.738–1.552] |
| | Allele | | | 0.00025 | 0.987 | 0.987 | |
| | G | 786 | 795 | | | | 1 |
| | A | 70 | 71 | | | | 1.997 [0.707–1.407] |
| rs12496815 | GG | 85 | 105 | 0.66 | 0.416 | 0.624 | 1 |
| | GA + AA | 303 | 327 | | | | 1.145 [0.826–1.586] |
| | Allele | | | | | | |
| | G | 347 | 423 | 2.953 | 0.086 | 0.172 | 1 |
| | A | 429 | 441 | | | | 1.186 [0.976–1.440] |

**Note:**
$P_{adj}$ represent P corrected by FDR.

These associations were found only in females, but not in males. In the male group, there was a difference between cases and controls in the genotype distribution of rs3806706 ($P = 0.02$); however, after adjusting for multiple testing, the difference was not significant ($P_{ajd} = 0.12$) (Table 4).

### LD and haplotype analysis

As shown in Fig. 1, the LD analysis of rs45536634 and rs12496815 in *CACNA2D2* showed that the D' values were equal to one in both female and male groups. According to the results of LD analysis, haplotype association analyses of rs45536634–rs12496815 were conducted in females and males respectively, and the results are shown in Table 5. Three common haplotypes were estimated to have a frequency > 1%, and haplotype AG was significantly associated with SCZ (OR = 1.91, $P_{adj} = 0.0096$) in females.

### Statistical power

The statistical power for rs45536634, rs12496815 and rs3806706 were 0.675–0.999, 0.872–0.999 and 0.878–0.999, respectively, if the OR varied from 1.4 to 2.0.

### DISCUSSION

The association between variants of a number of genes and SCZ has been reported in previous studies. To the best of our knowledge, this is the first report of a significant association between rs45536634 of *CACNA2D2* and SCZ in females of the Northeast Han Chinese population.

It is known that Ca2+ ion represents one of the most important second messengers in the brain and plays an essential role in neuronal development, synaptic transmission and
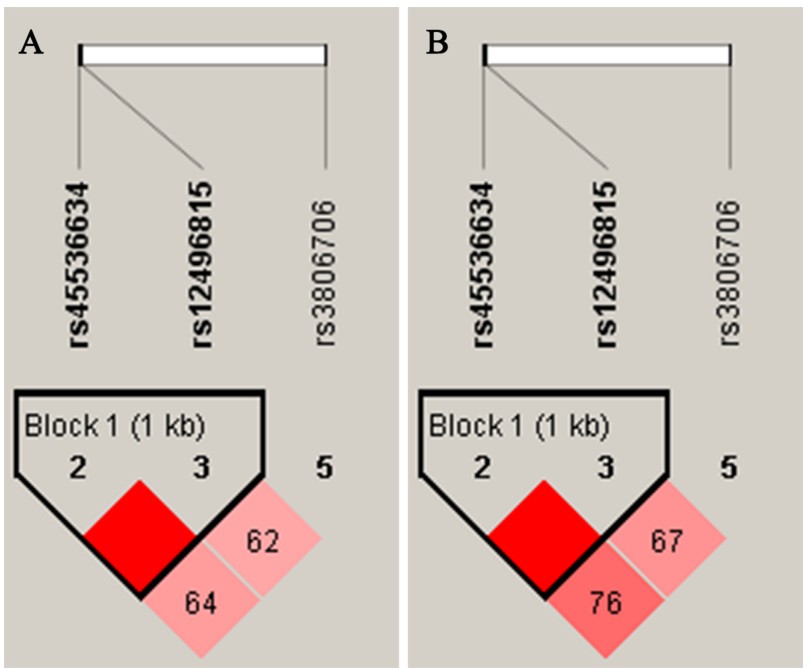

**Figure 1 Linkage disequilibrium (LD) of SNPs within *CACNA2D2* in female (A) and male (B) D' values were used toestimate the LD between pairwise SNPs.**

**Table 5 Association between haplotypes and schizophrenia by sex.**

| Haplotype | Male (frequency) | | | | | Female (frequency) | | | | |
|---|---|---|---|---|---|---|---|---|---|---|
| | Control | Case | OR [95% CI] | *P* | *P*$_{adj}$ | Control | Case | OR [95% CI] | *P* | *P*$_{adj}$ |
| rs45536634–rs12496815 | | | | | | | | | | |
| GA | 0.5105 | 0.5518 | 1 | – | – | 0.5424 | 0.5069 | 1 | – | – |
| GG | 0.4075 | 0.3666 | 0.84 [0.68–1.03] | 0.086 | 0.28 | 0.3942 | 0.3807 | 1.02 [0.80–1.31] | 0.87 | 0.87 |
| AG | 0.082 | 0.0816 | 0.93 [0.66–1.31] | 0.66 | 0.7 | 0.0634 | 0.1125 | 1.91 [1.26–2.90] | 0.0026 | 0.026* |
| AA | 0 | 0 | | – | – | 0 | 0 | | – | – |
| rs124996815–rs3806706 | | | | | | | | | | |
| GG | 0.4462 | 0.3933 | 1 | | | 0.4098 | 0.4196 | 1 | – | – |
| AG | 0.2659 | 0.2796 | 1.18 [0.92–1.51] | 0.18 | 0.3 | 0.2686 | 0.264 | 0.96 [0.71–1.29] | 0.77 | 0.87 |
| AC | 0.2439 | 0.2752 | 1.27 [1.00–1.61] | 0.046 | 0.23 | 0.2742 | 0.2473 | 0.88 [0.66–1.16] | 0.37 | 0.74 |
| GC | 0.044 | 0.0519 | 1.33 [0.77–2.29] | 0.31 | 0.388 | 0.0473 | 0.0691 | 1.43 [0.77–2.62] | 0.26 | 0.65 |
| rs124996815–rs3806706–rs45536634 | | | | | | | | | | |
| GGG | 0.3712 | 0.3148 | 1 | – | – | 0.3492 | 0.3297 | 1 | – | – |
| AGG | 0.266 | 0.2798 | 1.23 [0.95–1.59] | 0.12 | 0.28 | 0.2683 | 0.2598 | 1.04 [0.76–1.43] | 0.79 | 0.87 |
| ACG | 0.2439 | 0.2735 | 1.31 [1.03–1.69] | 0.031 | 0.23 | 0.2741 | 0.2496 | 0.97 [0.72–1.31] | 0.85 | 0.87 |
| GGA | 0.0749 | 0.0782 | 1.23 [0.83–1.81] | 0.3 | 0.288 | 0.061 | 0.0941 | 1.67 [1.02–2.72] | 0.042 | 0.167 |
| GCG | 0.0369 | 0.0502 | 1.58 [0.87–2.87] | 0.14 | 0.28 | 0.045 | 0.0482 | 1.18 [0.60–2.34] | 0.63 | 0.87 |
| rare | 0.0071 | 0.0035 | 0.72 [0.13–3.85] | 0.7 | 0.7 | 0.0024 | 0.0186 | 8.05 [1.01–64.35] | 0.05 | 0.167 |

**Note:**
* $P_{adj} < 0.05$.

plasticity, besides regulating various metabolic pathways (*Striessnig et al., 2006*). Notably, as demonstrated by several studies, the Ca2+ homeostasis disorder is associated with many pathological mechanisms, especially those related to neurodegenerative disorders, such as SCZ, Alzheimer's disease, and bipolar disorder (*Berridge, 2013*; *Sulzer & Surmeier, 2013*). *CACNA2D2* encodes the Alpha 2 delta 2 subunit of voltage-gated calcium channel (*Tedeschi et al., 2016*) which is a key signaling element, allowing changes in membrane potential to control a large number of Ca2+ dependent neurotransmitter release and neuronal plasticity in electrically excitable cells (*Striessnig et al., 2006*). A study conducted by *Villela et al. (2016)* showed that the copy number change in *CACNA2D2* was a risk factor for Alzheimer's disease.

Moreover, *CACNA1C*, a gene in the same family as *CACNA2D2*, has been repeatedly confirmed as one of the susceptibility genes for SCZ in various populations (*Guan et al., 2014*; *He et al., 2014*). In addition, *Zhang et al. (2018)* conducted a review research on calcium channel genes associated with SCZ in the Han Chinese population and found that *CACNA1C*, *CACNB2*, *CACNA2D1* and *CACNA2D3* were related to SCZ.

The sex-specific molecular phenotype of SCZ was observed in previous studies. A study conducted by Oumaima et al. indicated that minor alleles of SNPs in genes *LTA* and *TNFA* were over-represented in male SCZ patients but not in female schizrenia patients (*Inoubli et al., 2018*). *Jemli et al. (2017)* researched the association between the functional polymorphism of *IFNGR2* with SCZ and found the *IFNGR2 Q64R* polymorphism was associated with SCZ in males. The study conducted by *Yang et al. (2016)* showed that the genotypes and allele distributions of rs3087494 in *PLA2G12A* were significantly associated with SCZ in males, but not in females. Another study focused on sex-specific molecular phenotypes found that eight genes showed a differential expression in female and male schizophrenia patients (*Ramsey et al., 2013*).

For a better understanding of the association between SCZ and *CACNA2D2*, a more in-depth investigation—haplotype analysis was carried out to determine whether the combination of specific alleles was associated with the SCZ risk. The AG haplotype, consisted of rs44536634 and rs12496815 alleles, was correlated with an increased risk of SCZ in Han Chinese women. The haplotype analysis not only confirmed the association between rs44536634 and SCZ, but also supported that rs44536634 allele A was associated with an increased risk of SCZ in females. Furthermore, our research provided an evidence to support the distinct molecular phenotypes of SCZ patients with different gender, as reported in previous studies (*Ben Nejma et al., 2013*; *Jemli et al., 2017*; *Ramsey et al., 2013*).

In this study, several limitations should be considered. Firstly, our study was performed at a single center and only three SNPs were analyzed, it ignored SNPs in other genes that may be associated with SCZ. Secondly, the study is limited to interpreting the causal relationship between genetic risk factors and SCZ as this is a cross-sectional study. Furthermore, the representative of this study was limited to the adults of Northeast China because the samples were collected from Jilin Province. Finally, we lost some demographic covariates of the controls when analyzing the association between *CACNA2D2* SNP polymorphism and SCZ patients due to the difficulty of demographic characteristics

collection. Large-scale examination with more demographic characteristics is warranted to further examine the association between *CACNA2D2* and SCZ.

## CONCLUSION

The sample size of the present study was sufficient for detecting the effect of *CACNA2D2* variants on SCZ. This study demonstrated that *CACNA2D2* polymorphisms might influence the susceptibility to SCZ in Han Chinese women from Northeast China. The findings support the hypothesis that *CACNA2D2* may represent a novel susceptibility gene for SCZ in females. Functional genomics studies should be performed in future to validate the function of SCZ-associated *CACNA2D2* variants.

### Funding
This work was supported by the National Natural Science Foundation of China (No. 81673253). The funders had no role in study design, data collection and analysis, decision to publish, or preparation of the manuscript.

### Grant Disclosures
The following grant information was disclosed by the authors:
National Natural Science Foundation of China: 81673253.

### Competing Interests
The authors declare that they have no competing interests.

### Author Contributions
- Yingli Fu conceived and designed the experiments, performed the experiments, analyzed the data, prepared figures and/or tables, authored or reviewed drafts of the paper, and approved the final draft.
- Na Zhou performed the experiments, analyzed the data, prepared figures and/or tables, and approved the final draft.
- Wei Bai performed the experiments, prepared figures and/or tables, and approved the final draft.
- Yaoyao Sun performed the experiments, authored or reviewed drafts of the paper, and approved the final draft.
- Xin Chen performed the experiments, authored or reviewed drafts of the paper, and approved the final draft.
- Yueying Wang performed the experiments, authored or reviewed drafts of the paper, and approved the final draft.
- Mingyuan Zhang performed the experiments, authored or reviewed drafts of the paper, and approved the final draft.
- Changgui Kou conceived and designed the experiments, authored or reviewed drafts of the paper, and approved the final draft.

- Yaqin Yu conceived and designed the experiments, authored or reviewed drafts of the paper, and approved the final draft.
- Qiong Yu conceived and designed the experiments, authored or reviewed drafts of the paper, and approved the final draft.

## Human Ethics

The following information was supplied relating to ethical approvals (i.e., approving body and any reference numbers):

The Jilin University granted Ethical approval to carry out the study within its facilities (Ethical Applicagtion Ref: 2014-05-01).

## Data Availability

The raw measurements are available as a Supplemental File.

## Supplemental Information

Supplemental information for this article can be found online at http://dx.doi.org/10.7717/peerj.8521#supplemental-information.

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
