# Peer review of "Association of the CACNA2D2 gene with schizophrenia in Chinese Han population"

_PeerJ, doi:10.7717/peerj.8521_

## Round 0.1 · original submission · Major Revisions

I agree with our of our reviewers that "peer review is not a proof-reading service", so the authors should submit a clear and readable manuscript for peer review. Unfortunately, the paper is poorly written. This paper is very preliminary, please substantially revise it. Another round of extensive peer review is required.

Reviewer 1 ·

Basic reporting

no comment

Experimental design

no comment

Validity of the findings

no comment

Additional comments

Fu et al. performed an association analysis between 3 SNPs in CACNA2D2 and schizophrenia. The results showed that CACNA2D2 polymorphisms may influence the susceptibility to schizophrenia in female in Han Chinese population of Northeast China. However, there are a couple of fundamental issues that require further attention:
1. The authors should explain why do you choose the rs3806706, rs45536634 and rs12496815 as research objects. Have you checked these 3 SNPs in large scale GWASs (Li Z et al. 2017 Nat Genet, PGC 2014 Nature, Pardiñas et al. 2018 Nat Genet or others)?
2. In Statistical Analysis section, please describe which population was used to test the LD of these 3 SNPs.
3. You can also get the detailed information about this gene in SZDB2.0(www.szdb.org).
4. As shown in figure 1, the 3 SNPs were in linkage disequilibrium, especially between the rs45536634 and rs12496815. Why do you choose SPNs in LD?
5. Some typing errors should be corrected.

Reviewer 2 ·

Basic reporting

In this study, the authors performed a case-control study on the associations between SNPs in CACNA2D2 and Schizophrenia in Chinese Han population, they demonstrated the SNP rs45536634 showed significant association with schizophrenia and the haplotype AG consisting of rs45536634 and rs12496815 showed higher risk of schizophrenia.The article written clear. Relevant prior literatures were appropriately referenced.

Experimental design

1. Only three SNPs (rs3806706, rs45536634 and rs12496815) in CACNA2D2 were selected in this study, the authors should describe the reasons for selecting these SNPs, why the SNPs in exons and introns of CACNA2D2 were not selected, and whether the authors consider choosing the tag SNPs.
2. A total of 761 schizophrenia patients and 775 healthy controls were enrolled in this study. The authors should perform and describe the power calculation for each SNP according to the MAF of SNP and the sample size. If the statistical power is not enough, I suggest that the authors supplement the sample size.

Validity of the findings

The authors used the MALDI-TOF-MS method for SNP genotyping. This method is notoriously error prone and the authors should mention the accuracy of this method in their hands (repeated measurements, verification of SNPs by other methods).

Additional comments

The D’ value for establishing the haplotypes should be described in the Methods.

Reviewer 3 ·

Basic reporting

In this paper, Qiong Yu et al. studied the association between the Voltage-gated calcium channel CACNA2D2 and schizophrenia. By analyzing single-nucleotide polymorphisms (SNP) and haplotypes of CACNA2D2 in schizophrenia patients and healthy controls, they found one of these SNPs (rs45536634) significant associated with female schizophrenia patients. Furthermore, they claimed that people with haplotype AG consisting of rs45536634 and rs12496815 increased the risk of schizophrenia. The manuscript is poor-written, the experimental design is not so professional and the significance of this study is limited.

1. This manuscript is poorly written, with a multitude of English and grammatical errors. Using title as example “Polymorphism of CACNA2D2 and Association with Schizophrenia in a Han Chinese Population”. It should be “Association of the CACNA2D2 gene with schizophrenia in Chinese Han population”. Peer review is not a proof-reading service, and the authors should ensure that the manuscript has been passed through a professional grammatical proof reader if their own ability isn't sufficient to spot these errors.

Experimental design

2. The rationale for investigating the association of the CACNA2D2 gene with schizophrenia is not clear. 3p21 is not a schizophrenia risk area and CACNA2D2 is never mentioned in any GWAS of schizophrenia.

3. As the authors have already realized, the sample size of this study was not sufficient. Especially the positive association was only assessed from half of all the participants (female), which increased the concern that the association between of CACNA2D2 and schizophrenia susceptibility may be biased.

Validity of the findings

4. This study only analyzed three SNPs of CACNA2D2, which is insufficient and biased. Thus, the evidence that supports their conclusions are not so strong.

Reviewer 4 ·

Basic reporting

No comment

Experimental design

No comment

Validity of the findings

No comment

Additional comments

This paper presents the results of a genetic association study that aimed to identify the association between SNPs of CACNA2D2 gene and schizophrenia by a case-control study. In order to better improve the quality of the article, some issues should be solved. My detailed comments are as follows:

Title:
-There are some syntax errors in the title. Please correct them.

Abstract:
-Please add the purpose of the study in the “background” section.
-Please check the content of the abstract in the submission system. Some descriptions were omitted.

Introduction:
-Authors mentioned that showed SNPs of CACNA1C were strongly significant associated with schizophrenia and have been confirmed in different races. However, I suggest that authors should use more new literatures. For example,
1. Gassó P, Sánchez-Gistau V, Mas S, et al. Association of CACNA1C and SYNE1 in offspring of patients with psychiatric disorders[J]. Psychiatry research, 2016, 245: 427-435.
2. Zhang S Y, Hu Q, Tang T, et al. Role of CACNA1C gene polymorphisms and protein expressions in the pathogenesis of schizophrenia: a case-control study in a Chinese population[J]. Neurological Sciences, 2017, 38(8): 1393-1403.
3. Zhu X, Li R, Kang G, et al. CACNA1C Polymorphism (rs2283291) Is Associated with Schizophrenia in Chinese Males: A Case-Control Study[J]. Disease markers, 2019, 2019.
-Authors should add related research hypothesizes

Methods:
-Authors select three SNPs (rs3806706, rs45536634 and rs12496815) located in CACNA2D2 gene. I am very curious why the author chose 3 SNPs in so many SNPs of CACNA2D2 gene? Please explain the reason in detail.

Results:
-I think that authors should provide more demographic characteristics, such as education status, marital status and ect. in the result section.
-I suggest that authors should analyze genetic model and power test to avoid the false negative results.

Discussion:
-Line 178. “Yang Guang et.al showed that…” This is a wrong expression. Please correct it.

Conclusion:
-Authors did not describe any conclusion in the last end of the article

Reference:
-Line 279-281. Please re-check the format of this reference.

Table and Figure
-In Table 5, authors should list all possible haplotypes.

---

## Round 0.2 · Major Revisions

Thank you for your revisions. The paper seems much better than the previous version, however, it still has some issues that need to be addressed before its further process. Please consider these comments and revise it.

Reviewer 1 ·

Basic reporting

In this Major revision version, the authors revised the manuscript according to the suggestion of 4 reviewers. However, I didn't think this manuscript meets the requirements for publication in PeerJ. There are still many grammatical mistakes in the manuscript.

Experimental design

1. Almost all the reviewers raised the question "why did the authors select the three SNPs (rs3806706, rs45536634 and rs12496815) in CACNA2D2?". But I'm not covinced by the authors' response and the answer was not included in any part of the manuscript.
2. I'm also not satisfied with the answer "Why did you choose SNPs in LD?". It is a common sense to check LD of different SNPs before conducting experiments. Detecting SNPs in LD (rs45536634 and rs12496815) means you only get genotype information of one SNP.
3. Did you check the MAF of these 3 SNPs in different populations (CEU, CHS, YRI)?

Validity of the findings

no comment

Additional comments

Please correct the references "Consortium 2013" and "Consortium 2014" in Line 79 and Line 242.

Reviewer 2 ·

Basic reporting

The authors have addressed all of my concerns.

Experimental design

The authors have addressed all of my concerns.

Validity of the findings

The authors have addressed all of my concerns.

Additional comments

The authors have addressed all of my concerns.

Reviewer 3 ·

Basic reporting

Now, the authors adequately responded to the reviewer's concerns, and the manuscript is acceptable for publication.

Experimental design

N/A

Validity of the findings

N/A

Additional comments

N/A

Reviewer 4 ·

Basic reporting

None

Experimental design

None

Validity of the findings

None

Additional comments

The authors have made good responses to the comments of this reviewer.

---

## Round 0.3 · accepted · Accept

This revised version can be accepted now.

Reviewer 1 ·

Basic reporting

No comment.

Experimental design

No comment.

Validity of the findings

No comment.

Additional comments

The recently published East Asian GWAS of schizophrenia particularly highlighted a new-found locus - CACNA2D2. The authors can check whether the top association (rs374528934) was in LD with rs45536634. In all, the authors have adressed all my concerns.